# Preliminary Effects of Extended Reality-Based Rehabilitation on Gross Motor Function, Balance, and Psychosocial Health in Children with Cerebral Palsy

**DOI:** 10.3390/bioengineering12070779

**Published:** 2025-07-18

**Authors:** Onebin Lim, Yunhwan Kim, Chanhee Park

**Affiliations:** 1Department of Physical Therapy, Mokpo Science University, Mokpo 58644, Republic of Korea; onebin007@naver.com; 2Department of Physical Therapy, Yonsei University, Wonju 26493, Republic of Korea; aidendec8@gmail.com; 3Department of Physical Therapy, Jeonju University, Jeonju 55069, Republic of Korea

**Keywords:** extended reality, cerebral palsy, gross motor function, pediatric rehabilitation, parenting stress, quality of life

## Abstract

Extended reality (XR)-based rehabilitation is an emerging therapeutic approach that combines real and virtual environments to enhance patient engagement and promote motor and cognitive recovery. Its clinical utility in children with cerebral palsy (CP), particularly regarding gross motor skills, balance, and psychosocial well-being, remains underexplored. This preliminary study aimed to evaluate the potential effects of XR-based rehabilitation on gross motor function, balance, parental stress, and quality of life in children with cerebral palsy. Thirty children with cerebral palsy were randomly assigned to an extended reality training group (XRT, n = 15) or a conventional physical therapy group (CPT, n = 15). Both groups received 30 min sessions, three times per week for 6 weeks. Outcome measures included the Gross Motor Function Measure-88 (GMFM-88), Pediatric Balance Scale (PBS), Functional Independence Measure (FIM), Parenting Stress Index (PSI), and Pediatric Quality of Life Inventory (PedsQL), assessed pre- and post-intervention. A 2 (group) × 2 (time) mixed ANOVA was conducted. The XR group demonstrated improvements in GMFM-88, PBS, and FIM scores, with decreased PSI and increased PedsQL scores. Although most interaction effects were not statistically significant (GMFM-88: η^2^ = 0.035, *p* = 0.329; PBS: η^2^ = 0.043, *p* = 0.274), a marginal interaction effect was observed for PSI (*p* = 0.065, η^2^ = 0.059), suggesting a potential benefit of XR-based rehabilitation in reducing parental stress. This preliminary study indicates that XR-based rehabilitation may provide beneficial trends in motor function and psychosocial health in children with CP, particularly in reducing parental stress. Further studies with larger sample sizes are needed to confirm these findings.

## 1. Introduction

Cerebral palsy (CP) is a non-progressive neurodevelopmental disorder characterized by motor impairments and postural control deficits, often accompanied by cognitive, emotional, and sensory challenges [1]. These impairments significantly hinder the functional independence and quality of life in affected children, requiring consistent therapeutic interventions to improve gross motor function, balance, and psychosocial well-being [2,3].

Traditional rehabilitation approaches, including neurodevelopmental therapy and conventional physical therapy, have shown varying degrees of success in enhancing functional outcomes in children with CP [4]. However, these methods often lack engagement and motivation, especially in pediatric populations, limiting their long-term effectiveness [5].

Extended reality (XR), encompassing virtual reality (VR), augmented reality (AR), and mixed reality (MR), has emerged as an innovative tool in neurorehabilitation [6,7]. XR-based interventions allow children to interact with immersive environments that stimulate sensorimotor processing, promote repetitive task practice, and increase intrinsic motivation [8]. Previous studies have demonstrated the feasibility of XR in pediatric populations and its positive impact on balance, motor control, and emotional engagement. Nevertheless, empirical evidence on its clinical effectiveness in children with CP, particularly through controlled trials, remains limited [8,9]. For example, AlSaif and Alsenany (2015) applied virtual reality balance training in children with cerebral palsy and observed significant improvements in postural control and gross motor function [10]. Similarly, Liu et al. (2022) [11] conducted a meta-analysis of 16 trials involving 513 children and found that XR-based interventions significantly enhanced balance and gross motor function, with additional improvements in daily living ability. However, few studies have comprehensively evaluated psychosocial outcomes, such as motivation, social interaction, and emotional well-being, especially in the context of immersive XR rehabilitation. This study aims to address that gap [11].

In addition to motor outcomes, psychosocial factors such as parental stress and the child’s quality of life are critical yet often overlooked in rehabilitation studies. Parents of children with CP frequently report elevated stress levels due to caregiving demands, which may influence treatment adherence and the child’s overall development.

This preliminary study aims to investigate the effects of XR-based rehabilitation on gross motor function, balance, functional independence, parenting stress, and quality of life in children with CP, comparing it with conventional physical therapy. We hypothesize that XR-based intervention would demonstrate beneficial trends across motor and psychosocial domains, warranting further large-scale investigation.

## 2. Materials and Methods

### 2.1. Study Participants

This preliminary study employed a randomized controlled design involving 30 children diagnosed with cerebral palsy. Participants were recruited from a pediatric rehabilitation center and met the following inclusion criteria: (1) medical diagnosis of CP, (2) Gross Motor Function Classification System (GMFCS) levels I–III, (3) aged between 5 and 12 years, and (4) ability to follow verbal instructions. Children with uncontrolled epilepsy, severe visual/hearing impairments, or recent orthopedic surgery were excluded. Participants were randomly assigned to either the extended reality training group (XRT, n = 15) or the conventional physical therapy group (CPT, n = 15). This study was approved by the Daejeon Community Center Institutional Review Board (No. IRB-2022-DCC-06). Participants were recruited from a community welfare center located in Daejeon, Republic of Korea. This trial was not prospectively registered due to its preliminary pilot nature. The experiments were conducted by the ethical standards of the Committee on Human Experimentation of the institution in which the experiments were performed in accordance with the Declaration of Helsinki. Written informed consent was obtained from parents or guardians of the children. After the participants were recruited via bulletin board notices within the community center and presented to parents and children, an initial screening was conducted to determine whether the potential patients met the inclusion criteria. Informed consent was obtained from all patients before participation.

### 2.2. Experimental Procedure

All measurement instruments underwent daily calibration to ensure data accuracy, and a procedural checklist was employed to maintain consistency across experimental protocols. Prior to the initial outcome evaluations, participants were randomly allocated to either the XRT or CPT group via a coin-flip method during the first week following admission. The outcome measures comprised standardized assessments, including Activities of Daily Living (ADL) scores, Parenting Stress Index, quality of life, balance, and gross motor function. Assessments were administered at regular intervals, and participants completed a structured questionnaire capturing demographic and health-related information.

### 2.3. Outcome Measurements

Gross motor function was assessed using the Gross Motor Function Measure-88 (GMFM-88), a validated tool designed for children with cerebral palsy. Balance was evaluated with the Pediatric Balance Scale (PBS), which consists of 14 functional balance tasks. Functional independence was measured using the Functional Independence Measure (FIM), covering self-care, mobility, and communication domains. Parental stress was assessed using the Parenting Stress Index, Fourth Edition (PSI), and the Pediatric Quality of Life Inventory (PedsQL) was used to evaluate children’s health-related quality of life. All instruments have established validity and reliability in pediatric populations [12,13,14,15,16]. The primary outcome was gross motor function, as assessed by GMFM-88. All outcome assessments were performed by two independent assessors who were blinded to group allocation.

### 2.4. Intervention

Participants were randomly assigned to either the XRT or CPT group and received 30 min sessions, 4 days/week for 3 weeks. The CPT group received conventional neurodevelopmental treatment (NDT), focusing on mobility, balance, muscle strengthening, and stretching [5,17].

The XRT group received NDT once weekly and XR-based therapy three times weekly. XR interventions were customized according to the child’s GMFCS level, ensuring level-appropriate task complexity and motor demands [6]. Using MetaQuest 3 headsets, participants interacted with immersive virtual environments that promoted motor planning, spatial awareness, and real-time decision-making.

Figure 1 and Figure 2 illustrate the distinct XR contents developed for GMFCS levels III and II, respectively—highlighting level-specific scenarios such as cyclist avoidance for level III and dynamic dodging/tracking tasks for level II. The content was designed to enhance engagement and functional movement while providing real-time feedback. All sessions were supervised to ensure safety.

Real-time performance data were collected using the built-in inertial sensors of the head-mounted display (HMD) and the handheld joystick controllers. These sensors allowed for accurate recognition of gross motor gestures such as reaching, dodging, and lateral shifting. The XR system tracked spatial trajectories, movement smoothness, and task completion time. Although the XR data were not used for statistical comparison in this preliminary study, therapists used them during sessions to monitor progress, provide real-time feedback, and adjust task difficulty when needed.

The XR contents shown in Figure 1 and Figure 2 were independently developed by the research team specifically for this study, rather than being commercially available games. The content was designed to reflect task-oriented, functional training principles aligned with each participant’s GMFCS level. For example, the bike path scenario for level III participants involved obstacle avoidance and dynamic steering tasks to promote lower-limb control and postural adjustments. Meanwhile, the dodging and tracking scenario for level II emphasized multidirectional weight shifting, hand-eye coordination, and anticipatory movement. All game modules were developed using Unity 3D and implemented on the MetaQuest 3 platform with custom calibration for pediatric users.

All XR sessions were conducted by licensed physical therapists with over 5 years of pediatric rehabilitation experience and prior training in immersive XR systems. Prior to study initiation, the therapists underwent a 10 h orientation program to become proficient in system setup, calibration, and task supervision.

### 2.5. Statistical Analysis

Means and standard deviations of the results are discussed here. Assuming a normal distribution, the Kolmogorov–Smirnov test was used to assess every variable. Based on the findings of our pilot investigation, which indicated an effect size (eta squared, η^2^ = 0.5) and power (1-β = 0.8), a power analysis was conducted using G-Power software (version 3.1.9.7, Dusseldorf University, Dusseldorf, Germany) to establish the required sample size (N = 27). Thirty individuals were recruited, with a 10% probability of dropout. Variability between the sexes was determined using the chi-square test. A 2 (group: XRT vs. CPT) × 2 (time: pre vs. post) mixed-design ANOVA was used to analyze differences in outcome measures. Significance was set at *p* < 0.05. When a significant main effect was found in the mixed-design ANOVA, post hoc comparisons were performed with Bonferroni correction to adjust for multiple comparisons. In addition to mixed-design ANOVA, paired t-tests were conducted within each group to assess pre–post intervention changes. To further explore between-group differences, independent *t*-tests (or Mann–Whitney U tests) were conducted on the change scores (post–pre) for each outcome measure. If the normality assumption was violated, Wilcoxon signed-rank tests were used instead. Effect sizes for interaction effects were calculated using partial eta squared (η^2^). According to Cohen’s criteria, η^2^ values of 0.01, 0.06, and 0.14 were interpreted as small, medium, and large effects, respectively. SPSS software (version 26.0, SPSS, Chicago, IL, USA) was used for statistical analyses. A 0.05 *p*-value was determined. According to Cohen’s benchmarks, partial η^2^ values of 0.01, 0.06, and 0.14 represent small, medium, and large effects, respectively.

## 3. Results

### 3.1. Demographic Characteristics of Participants (N = 30)

All 30 participants completed the intervention without any dropouts or losses to follow up (Figure 3). All 30 participants completed the intervention protocol without adverse events. Table 1 presents the baseline demographic and clinical characteristics of the participants. There were no statistically significant differences between the XRT and CPT groups in terms of age, sex distribution, body height, body mass, or CP classification (all *p* > 0.05), indicating that the two groups were comparable prior to the intervention. There were no significant between-group differences in baseline characteristics, including GMFCS level distribution (*p* = 0.89), indicating comparability prior to the intervention. Most participants were classified as spastic diplegia, with 15% presenting with ataxic features.

### 3.2. Clinical Outcome Measurements

Both the XRT and CPT groups showed improvements in GMFM-88, PBS, FIM, PSI, and PedsQL scores following the 6-week intervention. The XR group demonstrated greater mean improvements across most outcome measures. However, statistical analysis revealed no significant interaction effects for GMFM-88, PBS, FIM, or PedsQL (all *p* > 0.05), indicating similar patterns of change over time in both groups (Table 2).

Notably, the PSI demonstrated a significant between-group difference favoring the XR group (*p* = 0.001). Furthermore, a marginal interaction effect was observed (*p* = 0.065), suggesting a trend toward greater parental stress reduction in the XR group compared to the CPT group (Table 3 and Figure 4).

To further contextualize the results, partial eta squared (η^2^) values were calculated to estimate effect sizes. Table 4 summarizes the F-values, significance levels, and effect sizes for the interaction effects (Time × Group) across outcome measures.

As shown in Table 4, the Summary of 2 (group) × 2 (time) mixed ANOVA results shows interaction effects (Time × Group) and corresponding partial η^2^ values for each outcome measure.

The interaction effect for PSI approached a medium effect size, reinforcing the clinical relevance of parental stress reduction despite the *p*-value being slightly above the threshold of significance.

An independent t-test comparing post-intervention PSI scores revealed a significant between-group difference (*p* = 0.001), while the mixed ANOVA showed a marginal time × group interaction effect (*p* = 0.065).

Despite the marginal interaction effect (*p* = 0.065), the partial η^2^ of 0.059 suggests a moderate effect size for PSI, indicating a potentially meaningful impact of XR intervention on parental stress.

## 4. Discussion

This preliminary study explored the effects of XR-based rehabilitation in children with cerebral palsy and identified potential benefits in both motor and psychosocial domains. While statistical significance was not achieved in most outcome measures, trends observed in GMFM-88, PBS, and FIM suggest that XR intervention may support improvements in gross motor function, balance, and independence. These results indicate that XR-based rehabilitation may yield promising outcomes, particularly in alleviating parenting stress, while providing comparable improvements in motor and functional outcomes relative to conventional therapy. These results indicate that XRT may yield promising outcomes, particularly in alleviating parenting stress, while providing comparable improvements in motor and functional outcomes relative to conventional therapy.

The most noteworthy finding of this study was the significant between-group difference in parenting stress, as measured by the PSI. Parents in the XR group reported lower stress levels post-intervention compared to those in the conventional therapy group. This may be attributed to enhanced engagement and motivation observed in children during XR sessions, potentially reducing caregiver burden and increasing perceived treatment effectiveness [18,19]. The marginal interaction effect further suggests that XR rehabilitation may have a unique impact on parental well-being. Informal observations by therapists during the intervention sessions indicated that children in the XR group displayed greater interest and motivation compared to those in the CPT group. Children frequently expressed enjoyment and anticipation for XR sessions and were more engaged in task completion. This heightened engagement may have contributed to the observed reduction in parental stress, as caregivers often noted increased enthusiasm and cooperation from their children. While not quantitatively assessed in this study, these behavioral trends underscore the potential of XR-based therapy to enhance intrinsic motivation and adherence in pediatric rehabilitation. Although improvements in motor outcomes did not reach statistical significance, these findings are consistent with previous literature reporting favorable clinical trends with immersive and interactive rehabilitation in pediatric populations [20,21]. Effect size analyses revealed a partial η^2^ of 0.059 for PSI, indicating a medium effect, which supports the clinical relevance of stress reduction even in the absence of statistical significance. For other outcomes, the effect sizes were small (η^2^ < 0.05), reflecting limited group-by-time differences in physical function changes. This underscores the challenge of detecting robust treatment effects in small samples with short intervention durations [22]. The relatively limited statistical significance across outcomes may stem from several factors. First, the sample size (n = 30) was modest, which inherently reduces the power to detect interaction effects. Second, the intervention period of six weeks, though clinically feasible, may be insufficient for producing measurable functional changes in children with CP. Third, outcome measures such as GMFM and PBS, while widely validated, may require longer-term follow-up or more intensive dosage to reflect group-level differences in intervention response. Nevertheless, the present study contributes important preliminary evidence for the feasibility and potential benefit of XR-based rehabilitation, especially in psychosocial domains. The reduction in parenting stress is particularly noteworthy, as caregiver well-being directly influences treatment adherence and developmental outcomes. Additional outcome measures with higher sensitivity or domain specificity may be warranted in future studies. Moreover, inclusion of long-term follow-up or qualitative evaluations could provide a deeper understanding of the intervention’s sustained effects. Furthermore, the high level of participant retention and absence of adverse events support the acceptability and safety of XR interventions in pediatric rehabilitation settings. This study has several limitations. Besides the small sample size and short intervention period, the lack of IRB approval limits generalizability. Although the study was conducted ethically with informed consent, future trials should seek institutional oversight. Additionally, the absence of qualitative data from parents or children precludes deeper insight into subjective experiences and satisfaction. Future studies should consider larger, multi-center trials with longer duration, and include mixed-method approaches that capture both quantitative outcomes and qualitative experiences. Exploring personalized XR content, caregiver involvement, and gamified progression could further enhance the intervention’s therapeutic potential. Incorporating qualitative methods such as structured interviews or satisfaction surveys with children and their parents could yield valuable insights into user experience and inform further refinement of the XR program. Despite these limitations, the current findings contribute to the growing body of evidence supporting the application of XR in pediatric neurorehabilitation. XR-based therapy may serve as a promising adjunct to traditional rehabilitation by fostering motivation, engagement, and family-centered benefits. Continued exploration of XR in pediatric populations is warranted. Given the limited sample size and 6-week duration, future studies should consider extended intervention periods with larger cohorts to enhance statistical power and increase the generalizability of findings.

Future interventions should explore adaptive XR systems with real-time performance analytics, task customization, and biofeedback integration to optimize therapeutic impact. Moreover, standardized XR protocols could help improve replicability and cross-site comparability in multi-center trials. Future XR applications may benefit from adaptive design frameworks that tailor task complexity and content themes to individual user profiles, potentially enhancing engagement and therapeutic relevance. Involving caregivers in this design process may further optimize personalization.

## 5. Conclusions

This preliminary study found that XR-based rehabilitation may help reduce parenting stress and may offer motor function benefits comparable to conventional physical therapy. While most outcomes did not reach statistical significance, the intervention was safe, feasible, and well tolerated. Future large-scale studies are warranted.

## Figures and Tables

**Figure 1 bioengineering-12-00779-f001:**
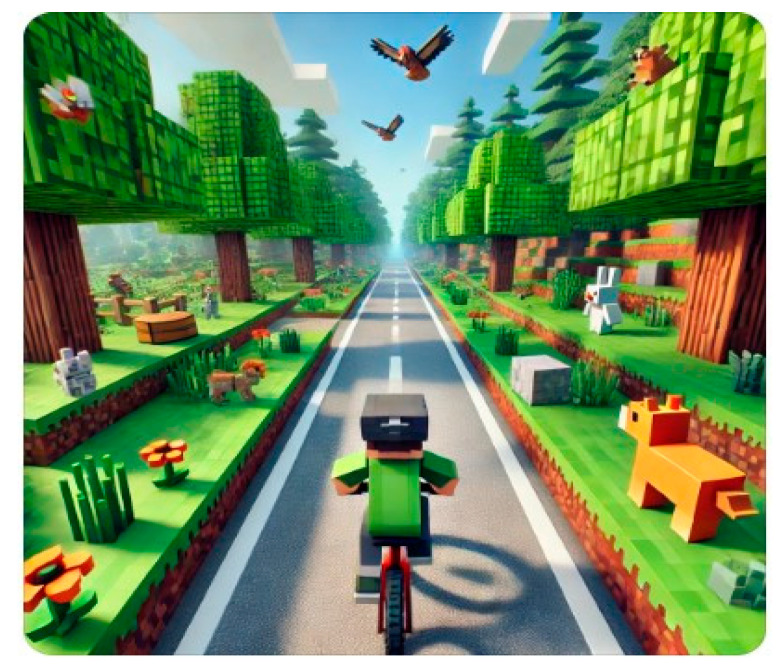
XR content for GMFCS level III participants: bike path scenario.

**Figure 2 bioengineering-12-00779-f002:**
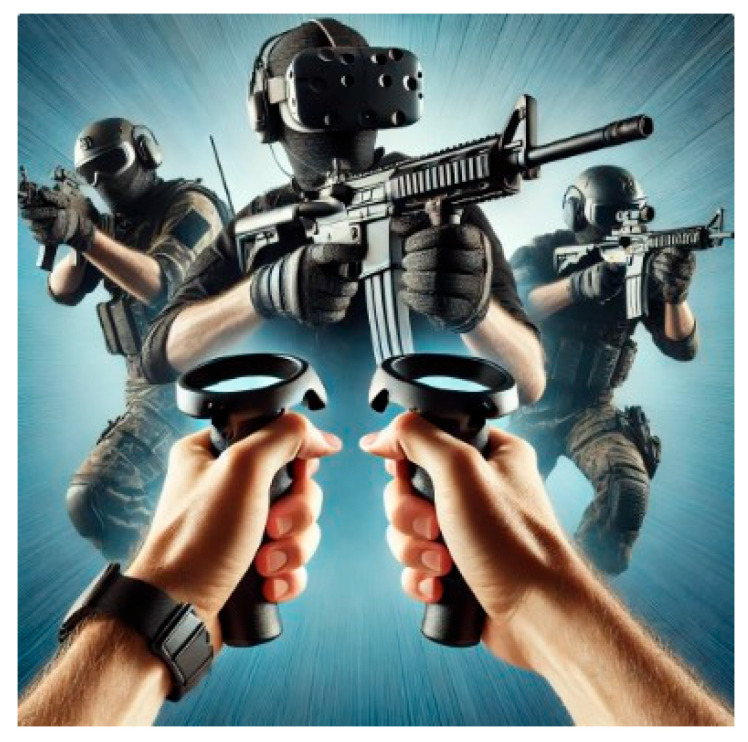
XR content for GMFCS level II participants: dodging and tracking game.

**Figure 3 bioengineering-12-00779-f003:**
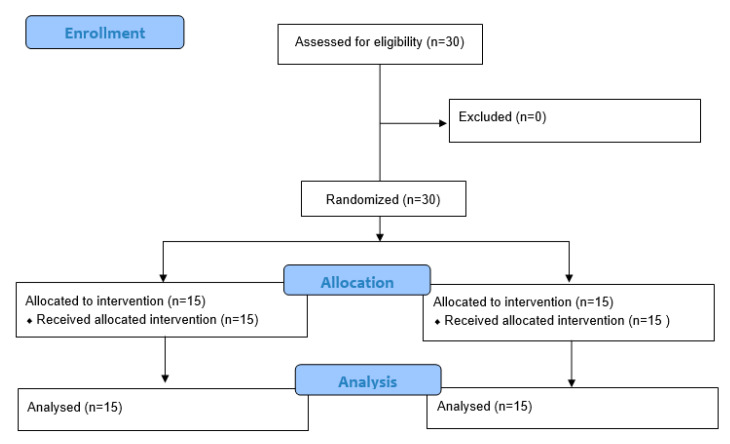
Flow chart.

**Figure 4 bioengineering-12-00779-f004:**
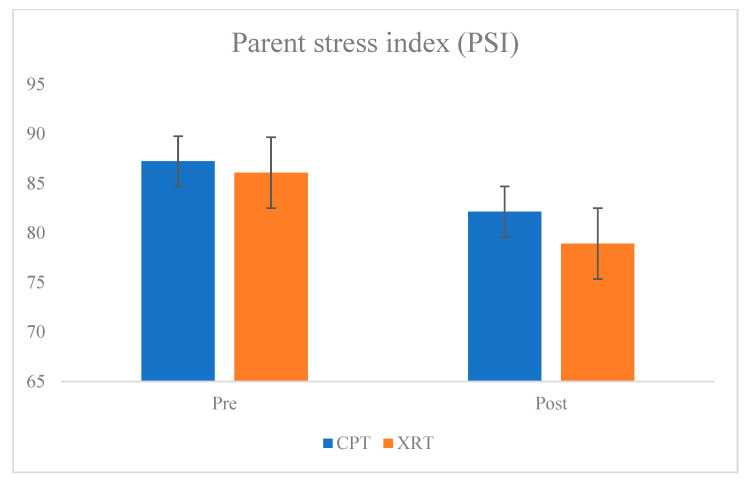
Parent stress index graph.

**Table 1 bioengineering-12-00779-t001:** Summarizes the demographic and clinical profiles of the participants.

	CPT (n = 15)	XRT (n = 15)	*p*-Value
Sex (male/female)	10/5	12/3	0.41
Age (years)	14.23 ± 2.88	15.42 ± 3.16	0.35
Body height (cm)	128.63 ± 19.27	129.6 ± 14.23	0.25
Body mass (kg)	29.13 ± 14.08	31.43 ± 10.76	0.36
GMFCS (I/II/III) ^a^	4/6/5	5/5/5	0.89

All analyses were based on N = 15 participants per group. ^a^ GMFCS: Gross Motor Function Classification System.

**Table 2 bioengineering-12-00779-t002:** Physical activity measurements.

	CPT			XRT			*p*-Value
	Pre-Test	Post-Test	With Group (Effect Size)	Pre-Test	Post-Test	With Group (Effect Size)	Time × Group
GMFM 88 ^a^	67.15 ± 18.42	71.12 ± 16.88	0.012 * (0.22)	68.08 ± 17.78	72.83 ± 15.82	0.018 * (0.28)	0.565
PBS ^b^	28.12 ± 12.56	28.88 ± 10.23	0.04 * (0.1)	16.68 ± 14.23	25.11 ± 8.88	0.001 * (0.71)	0.092
FIM ^c^	20.17 ± 8.21	20.90 ± 8.12	0.03 * (0.1)	20.25 ± 5.21	21.02 ± 7.23	0.04 * (0.1)	0.095

^a^ GMFM 88: Gross motor functional measure 88. ^b^ PBS: Pediatrics balance scale. ^c^ FIM: Functional Independence Measure. All analyses were based on N = 15 participants per group. * *p* < 0.05.

**Table 3 bioengineering-12-00779-t003:** Psychosocial measurements.

	CPT			XRT			*p*-Value
	Pre-Test	Post-Test	With Group (Effect Size)	Pre-Test	Post-Test	With Group (Effect Size)	Time × Group
PSI ^a^	87.23 ± 16.23	82.15 ± 14.88	0.01 * (0.3)	86.08 ± 14.83	78.92 ± 10.83	0.001 * (0.55)	0.107
PedsQL ^b^	50.25 ± 18.21	44.80 ± 15.83	0.02 * (0.32)	50.89 ± 16.22	44.48 ± 17.26	0.01 * (0.38)	0.14

^a^ PSI: Parent stress index, fourth edition. ^b^ PedsQL: Pediatric quality of life. * *p* < 0.05. All analyses were based on N = 15 participants per group.

**Table 4 bioengineering-12-00779-t004:** Summary of 2 × 2 mixed ANOVA interaction effect and effect size.

Outcome Measure	F (Time × Group)	*p*-Value	Partial η^2^
GMFM-88 ^a^	0.99	0.329	0.035
PBS ^b^	1.24	0.274	0.043
FIM ^c^	1.05	0.315	0.037
PSI ^d^	3.76	0.065	0.059
PedsQL ^e^	0.79	0.381	0.029

^a^ GMFM 88: Gross motor functional measure 88. ^b^ PBS: Pediatrics balance scale. ^c^ FIM: Functional Independence Measure. ^d^ PSI: Parent stress index, fourth edition. ^e^ PedsQL: Pediatric quality of life. All analyses were based on N = 15 participants per group.

## Data Availability

The original contributions presented in this study are included in the article. Further inquiries can be directed to the corresponding author.

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
