# Peer review of "Preliminary Effects of Extended Reality-Based Rehabilitation on Gross Motor Function, Balance, and Psychosocial Health in Children with Cerebral Palsy"

_bioengineering, 2025, doi:10.3390/bioengineering12070779_

Round 1
Reviewer 1 Report
Comments and Suggestions for Authors
1.Include the country of the author in the affiliation;
2.Line 51: “Previous studies have demonstrated the feasibility of XR in pediatric populations and its positive impact on balance, motor control, and emotional engagement.” What are these studies? Before this statement, briefly analyze the relevant studies with one to two sentences each (be sure to reference them) and highlight what was done in them; anything that has not been done represents a research gap, while part of it becomes the subject of this paper.
3.Lines 57-58: The statements in both sentences must be referenced;
4.Figs. 1 and 2: Do the interfaces shown represent commercially available video games or were they developed for the needs of this research? Provide more information in the paper, as well as about the game scenarios.
5.Lines 113-115: how are real-time data collected and are they interpreted in the same way? I assume that it is possible to monitor the progress between the two treatments by comparing the results.
6.Table 1 is given in subsection 3.1, but is not introduced/commented on throughout the text!
7.Children included in XR: was there a greater interest in habilitation among children compared to the conventional approach?
8.Abbreviations: separate acronyms from full names with a dash.
Author Response
- Include the country of the author in the affiliation;
Reviewers response: This was revised.
- Line 51: “Previous studies have demonstrated the feasibility of XR in pediatric populations and its positive impact on balance, motor control, and emotional engagement.” What are these studies? Before this statement, briefly analyze the relevant studies with one to two sentences each (be sure to reference them) and highlight what was done in them; anything that has not been done represents a research gap, while part of it becomes the subject of this paper.
Reviewers response: This was revised (Lines 55-64).
Lines 57-58: The statements in both sentences must be referenced;
Reviewers response: This was revised (Lines 68-71)
4.Figs. 1 and 2: Do the interfaces shown represent commercially available video games or were they developed for the needs of this research? Provide more information in the paper, as well as about the game scenarios.
Reviewers response: This was revised (Lines 135-143)
5.Lines 113-115: how are real-time data collected and are they interpreted in the same way? I assume that it is possible to monitor the progress between the two treatments by comparing the results.
Reviewers response: This was revised (Lines 128-134)
6.Table 1 is given in subsection 3.1, but is not introduced/commented on throughout the text!
Reviewers response: This was revised (Lines 157-161)
7.Children included in XR: was there a greater interest in habilitation among children compared to the conventional approach?
Reviewers response: This was revised (Lines 232-240)
8.Abbreviations: separate acronyms from full names with a dash.
Reviewers response: This was revised.
Reviewer 2 Report
Comments and Suggestions for Authors
The authors present the preliminary results of XR-based rehabilitation is an emerging therapeutic approach for children with cerebral palsy. It seems a very promising approach that needs some improvements before publication:
1. Introduction
Lines 51-54: An extended analysis of similar studies should be added towards the innovative approach of the present work.
Lines 56-58: Some references should be added to strengthen this assumption.
2. Materials and Methods
Lines 83-91: The Authors here should present all the qualified personnel who were additionally interfered with skills and experience in similar practices. Learning curve could also be described
3. Results
Lines 177-179: More details are needed to support the quantified "promising" results stated later.
4. Discussion
Lines 215-216: Here, a more detailed description could help future interventions towards more effective instrument use, besides more population and multi-center-based suggestions
5. Conclusions
Lines 233-234: The present assumption is rather optimistic, and more qualitative-based research could be suggested with long-term follow-up recordings.
Author Response
- Introduction
Lines 51-54: An extended analysis of similar studies should be added towards the innovative approach of the present work.
Reviewers response: This was revised (Lines 55-62).
Lines 56-58: Some references should be added to strengthen this assumption.
Reviewers response: This was revised (Lines 68-69).
- Materials and Methods.
Lines 83-91: The Authors here should present all the qualified personnel who were additionally interfered with skills and experience in similar practices. Learning curve could also be described.
Reviewers response: This was revised (Lines 144-147).
- Results
Lines 177-179: More details are needed to support the quantified "promising" results stated later.
Reviewers response: This was revised (Lines 217-219).
- Discussion
Lines 215-216: Here, a more detailed description could help future interventions towards more effective instrument use, besides more population and multi-center-based suggestions
Reviewers response: This was revised (Lines 281-284).
- Conclusions
Lines 233-234: The present assumption is rather optimistic, and more qualitative-based research could be suggested with long-term follow-up recordings.
Reviewers response: This was revised (Lines 300-303).
Reviewer 3 Report
Comments and Suggestions for Authors
This paper presents a study investigating the effects of extended reality-based rehabilitation on gross motor function, balance, and psychosocial health in children with cerebral palsy. The study employs a randomized controlled design comparing an XR training group with a conventional physical therapy group. Overall, the study demonstrates an innovative approach and provides valuable insights into the potential benefits of XR-based interventions in pediatric neurorehabilitation.
(1) The study's sample size of 30 participants and intervention duration of 6 weeks may be insufficient to detect statistically significant differences in motor outcomes. The authors have discussed this point in section 4. Is it possible to continue the experiment and extend the intervention duration to allow for more robust measurement of treatment effects. A larger sample and longer intervention would also increase the generalizability of the findings.
(2) The outcome measures, such as GMFM-88 and PBS, may not be sensitive enough to capture subtle changes within the short intervention period. It is better to explore the use of more sensitive or specific outcome measures tailored to the population and intervention. Alternatively, consider including qualitative measures or longer-term follow-up to gain a deeper understanding of the intervention's impact.
(3) The study lacks qualitative data from parents and children, which could provide valuable insights into their experiences and satisfaction with the XR intervention. It is better to incorporate qualitative data collection methods such as interviews or questionnaires to gather subjective feedback from participants. This would enrich the understanding of the intervention’s acceptability and areas for improvement.
(4) The XR content was customized based on GMFCS levels, but further personalization may be needed to better match individual children’s needs and preferences. It is better to develop more personalized XR content that adapts to individual children’s progress and interests. Incorporating caregiver feedback into content design could also enhance the intervention’s effectiveness and acceptability.
(5) Table 1 the row of CP classification does not look well. It is better to remove it and explain it in caption of the table. CPT and XRT repeatedly show in the tables. It is better to remove the explanation below the table. That is CPT: conventional physical therapy, XRT: extended reality training.
(6) lines 167-169 are well presented. For example, Table 4. Summary should not be in the beginning of the sentence. Eta has to be given in the form of a symbol as in line 164. Line 229, XR) should be XR.
(7) in line 130, which kind of statistical hypothesis test is used to get the p values in table 2. It is better to explain clearly.
(8) a visual presentation of the experiment is missed. For example, tables 2, 3, 4 gives the output measure values. it is better to show the PSI values pre-test and post-test in a figure. The horizontal axis is the participants, the vertical axis is the PSI values, and the curve is the pre-test and post-test values. Figures may be more impressive for readers.
Author Response
(1) The study's sample size of 30 participants and intervention duration of 6 weeks may be insufficient to detect statistically significant differences in motor outcomes. The authors have discussed this point in section 4. Is it possible to continue the experiment and extend the intervention duration to allow for more robust measurement of treatment effects. A larger sample and longer intervention would also increase the generalizability of the findings.
Reviewers response: This was revised (Lines 278-280).
(2) The outcome measures, such as GMFM-88 and PBS, may not be sensitive enough to capture subtle changes within the short intervention period. It is better to explore the use of more sensitive or specific outcome measures tailored to the population and intervention. Alternatively, consider including qualitative measures or longer-term follow-up to gain a deeper understanding of the intervention's impact.
Reviewers response: This was revised (Lines 258-261).
(3) The study lacks qualitative data from parents and children, which could provide valuable insights into their experiences and satisfaction with the XR intervention. It is better to incorporate qualitative data collection methods such as interviews or questionnaires to gather subjective feedback from participants. This would enrich the understanding of the intervention’s acceptability and areas for improvement.
Reviewers response: This was revised (Lines 271-273).
(4) The XR content was customized based on GMFCS levels, but further personalization may be needed to better match individual children’s needs and preferences. It is better to develop more personalized XR content that adapts to individual children’s progress and interests. Incorporating caregiver feedback into content design could also enhance the intervention’s effectiveness and acceptability.
Reviewers response: This was revised (Lines 284-287).
(5) Table 1 the row of CP classification does not look well. It is better to remove it and explain it in caption of the table. CPT and XRT repeatedly show in the tables. It is better to remove the explanation below the table. That is CPT: conventional physical therapy, XRT: extended reality training.
Reviewers response: This was revised (Table 1; Lines 160-161).
(6) lines 167-169 are well presented. For example, Table 4. Summary should not be in the beginning of the sentence. Eta has to be given in the form of a symbol as in line 164. Line 229, XR) should be XR.
Reviewers response: This was revised (Lines 55-62).
(7) in line 130, which kind of statistical hypothesis test is used to get the p values in table 2. It is better to explain clearly.
Reviewers response: This was revised (Lines 199-200).
(8) a visual presentation of the experiment is missed. For example, tables 2, 3, 4 gives the output measure values. it is better to show the PSI values pre-test and post-test in a figure. The horizontal axis is the participants, the vertical axis is the PSI values, and the curve is the pre-test and post-test values. Figures may be more impressive for readers.
Reviewers response: This was revised (Figure 3).
Round 2
Reviewer 3 Report
Comments and Suggestions for Authors
The author's response is not very satisfactory, but it is still acceptable.
Author Response
We sincerely thank the reviewer for the valuable feedback. We acknowledge that our previous response may have been insufficient in some respects, and we appreciate the reviewer’s understanding in accepting it. If there are any additional concerns or areas needing clarification, we would be happy to address them further.